# Molecular and Physiological Responses of Rice and Weedy Rice to Heat and Drought Stress †

Leonard Bonilha Piveta [1], Nilda Roma-Burgos [2,*], José Alberto Noldin [3], Vívian Ebeling Viana [1], Claudia de Oliveira [1], Fabiane Pinto Lamego [4] and Luis Antonio de Avila [1,*]

[1] Crop Protection Graduate Program (Programa de Pós-Graduação em Fitossanidade),
Federal University of Pelotas (Universidade Federal de Pelotas), Pelotas, RS CEP 96160-000, Brazil;
leonard_piveta@hotmail.com (L.B.P.); vih.viana@gmail.com (V.E.V.); Claudia.Oliveira@syngenta.com (C.d.O.)

[2] Department of Crop, Soil and Environmental Sciences, University of Arkansas, Fayetteville, AR 72704, USA

[3] Epagri/Itajai Experimental Station, Itajaí, SC 88318-112, Brazil; noldin@epagri.sc.gov.br

[4] Embrapa Pecuária Sul, Bagé, RS 96401-970, Brazil; fabilamego@yahoo.com.br

* Correspondence: nburgos@uark.edu (N.R.-B.); laavilabr@gmail.com (L.A.d.A.)

† This work was part of the Ph.D. dissertation of the first author.

**Abstract:** Rice is the staple food for about half of the world population. Rice grain yield and quality are affected by climatic changes. Arguably, rice cultivars' genetic diversity is diminished from decades of breeding using narrow germplasm, requiring introgressions from other *Oryza* species, weedy or wild. Weedy rice has high genetic diversity, which is an essential resource for rice crop improvement. Here, we analyzed the phenotypic, physiological, and molecular profiles of two rice cultivars (IRGA 424 and SCS119 Rubi) and five weedy rice (WR), from five different Brazilian regions, in response to heat and drought stress. Drought and heat stress affected the phenotype and photosynthetic parameters in different ways in rice and WR genotypes. A WR from Northern Brazil yielded better under heat stress than the non-stressed check. Drought stress upregulated *HSF7A* while heat stress upregulated *HSF2a*. *HSP74.8, HSP80.2,* and *HSP24.1* were upregulated in both conditions. Based on all evaluated traits, we hypothesized that in drought conditions increasing *HSFA7* expression is related to tiller number and that increase WUE (water use efficiency) and *HSFA2a* expression are associated with yield. In heat conditions, $G_s$ (stomatal conductance) and E's increases may be related to plant height; tiller number is inversely associated with *HSPs* expression, and chlorophyll content and $C_i$ (intercellular $CO_2$ concentration) may be related to yield. Based on morphology, physiology, and gene regulation in heat and drought stress, we can discriminate genotypes that perform well under these stress conditions and utilize such genotypes as a source of genetic diversity for rice breeding.

**Keywords:** genetic diversity; photosynthesis; heat shock proteins

## 1. Introduction

The demand for food increases with the increase in population and buying power of densely populated countries [1,2]. Agriculture has the challenge to meet this demand. The efficient use of natural resources, reducing waste, and increasing crop production, are essential to meet the growing food demand worldwide [3,4]. The challenge is to minimize soil degradation, use less water, and maintain ecological biodiversity in a changing climate where the agroecosystems are exposed to increasing abiotic stresses [4–6].

Rice is one of the world's staple cereals and is responsible for feeding nearly half of the world population [7], but rice is highly susceptible to changing climate and associated stresses like water shortage for irrigation [7,8]. In addition to abiotic stresses, weed competition also dampens rice production. Weedy rice, also called red rice, is considered one of the main weeds in rice areas in southern Brazil [9–13]. Weedy rice also infests rice fields in most paddy rice production regions globally [14–18], causing significant yield losses.

Despite its negative effect on rice yield and quality, weedy rice is a potential source of genetic diversity and desirable adaptive traits for rice breeding programs [19].

Environmental changes lead to a dynamic process of quick responses in plants [20]. A highly complex signaling system acts on reprogramming gene expression, mainly of stress-responsive genes [20–22]. Overall, the stress responses lead to a series of molecular, physiological, biochemical, and morphological changes [23,24]. Climate changes have been linked to the high frequency and severity of abiotic stress incidents [25]. High temperature and water deficit (drought) are two of the main factors that affect plant growth [26]. Cell damage, inhibition of photosynthesis, osmotic adjustment, induction of repair systems and chaperones, changes in gene expression, and metabolism are the general plant responses to these stresses [27]. Products of genes induced by stress are classified into two major groups: (1) proteins that directly protect the cell against stress, including chaperones, late embryogenesis abundant (LEA) proteins, antifreeze proteins, and detoxification enzymes; and (2) proteins that regulate gene expression and signal transduction pathways such as transcription factors (TFs) [22].

Heat shock proteins (HSPs) are molecular chaperones that regulate the protein folding, localization, accumulation, and degradation in different species of eukaryotes and prokaryotes, playing a wide array of roles in many cellular processes, helping in cell homeostasis and plant survival in adverse conditions [28]. HSPs are classified into five families according to their molecular weights: HSP100, HSP90, HSP70, chaperonins or HSP60 and low molecular weight HSPs (12–40 KDa) named small HSPs (sHSPs) [29]. HSPs present a constitutive or inducible expression pattern. Constitutively expressed proteins (under favorable environmental conditions) act as chaperones or proteases, displaying numerous intracellular functions. Chaperones are involved in protein synthesis, folding, and transport, while proteases act on the degradation of damaged proteins [30]. Together, chaperones and proteases minimize the chance of inappropriate protein interactions, which can result in harmful changes in certain organisms [28]. In contrast, inducible HSPs are largely synthesized under stress conditions in response to minimize or repair cell damage, ensuring the survival of the cell or organism, and/or inducing tolerance to subsequent stresses, such as high temperature [31] and drought [32].

The transcription of most genes encoding HSPs is regulated by heat shock TFs (HSFs) located in the cytoplasm [33,34]. HSFs are the final components of the signal transduction pathway that mediate the activation of heat-responsive genes and many chemical stressors [35]. Plants overexpressing genes coding for HSFs display increased tolerance to heat, salt, osmotic, drought, and cold stress [34,36–38]. Expression of *HSFs* in response to different abiotic stresses indicates that these TFs can regulate multiple mechanisms. Thus, HSFs are considered a vital gene family, and their potential to respond to different stresses can be used to obtain plants that are tolerant to environmental stresses [36]. First identified in tomato, the HSF family was characterized in a wide range of species, including *Arabidopsis thaliana*, rice, and maize, wherein 21, 25, and 25 genes were identified in each species, respectively [39–43]. HSFs are the first responsive molecules involved in cell stress signaling and gene expression activation [44]. Under normal growth conditions, chaperones maintain the inactive state of the HSFs and in response to stress, HSFs dissociate from chaperones [45].

Given the importance of HSPs for thermotolerance, it is often assumed that stressed-organisms accumulate higher levels of HSPs [27,46]. On the other hand, some studies demonstrate that the expression of HSPs can be downregulated in stressful environments [47,48]. Overall, this study aimed to evaluate changes in morphology, photosynthetic capacity, and gene expression of *HSPs* and *HSFs* in rice cultivars and weedy rice genotypes under drought and heat stress.

## 2. Material and Methods

### 2.1. Plant Material

The experiment was performed in the growth chambers at the Altheimer Laboratory, Department of Crop, Soil, and Environmental Sciences (CSES), University of Arkansas, Fayetteville, AR, USA, in 2016–2017. The genotypes evaluated were two rice cultivars (*Oryza sativa* L.) and five weedy rice (WR) accessions (Table 1). The commercial rice cultivars were IRGA 424 (white rice) and SCS119 Rubi (red rice) [49]. Weedy rice seeds were obtained from three geographic regions in Brazil: (1) South (Rio Grande do Sul, RS, and Santa Catarina, SC) (2) Northeastern (Rio Grande do Norte, RN, and Paraíba, PB); and (3) North (Roraima, RR) (Table 1, Figure S1). The different regions have different rainfall and mean temperature patterns throughout the year (Figure 1). The weedy rice accessions were collected in 2012/2013 from commercial rice fields. The seeds were multiplied for three generations to homogenize the populations by removing off-type plants and increasing the number of seeds (Arroio Grande, RS; Capão do Leão, RS; Fayetteville, AR, USA, 1st, second and third generation, respectively).

**Table 1.** Geographic origin of rice cultivars and weedy rice accessions used in the study.

| Cultivar/Biotype | City/State | Brazil Region | GPS Coordinates | |
| --- | --- | --- | --- | --- |
| | | | Latitude | Longitude |
| IRGA 424 | Pelotas, RS | South | 31°46′19″ S | 52°20′33″ W |
| SCS119 Rubi | Itajaí, SC | South | 26°54′28″ S | 48°39′43″ W |
| WR_SC | Gaspar, SC | South | 26°54′41″ S | 48°50′60″ W |
| WR_RS | Dom Pedrito, RS | South | 31°02′07″ S | 54°52′02″ W |
| WR_PB | São José do Rio do Peixe, PB | Northeastern | 6°44′11″ S | 38°26′39″ W |
| WR_RN | Apodi, RN | Northeastern | 5°44′55″ S | 37°46′27″ W |
| WR_RR | Bonfim, RR | North | 3°19′56″ N | 59°54′18″ W |

The soil was collected from a research field at the University of Arkansas Milo Shult Agricultural Research & Extension Center, Fayetteville, AR, USA. The soil was classified as Captina silt-loam, with the following characteristics: sand = 30.5%; silt = 55.5%; clay = 14%, $pH_{water} = 7.3$; organic matter = 2.41%, $NO_3 = 32.4$ mg kg$^{-1}$, $NH_4 = 16.8$ mg kg$^{-1}$, P = 86 mg kg$^{-1}$, K = 41 mg kg$^{-1}$, Ca = 827 mg kg$^{-1}$, Mg = 827 mg kg$^{-1}$, S = 10 mg kg$^{-1}$, Na = 22 mg kg$^{-1}$, Fe = 671 mg kg$^{-1}$, Mn = 168 mg kg$^{-1}$, Zn = 3.6 mg kg$^{-1}$, Cu = 0.6 mg kg$^{-1}$, B = 0.2 mg kg$^{-1}$. The experimental unit was a 1-L pot filled with soil, planted with rice or weedy rice genotypes, arranged in a completely randomized design with three replicates. Five seeds were sown per pot and later thinned to one seedling per pot.

### 2.2. Drought and Heat Stress Induction

The stress treatments were imposed during the vegetative stage of rice and weedy rice plants. The experiment was performed under a 14-h photoperiod and temperature of 25 °C (night) to 30 °C (day). The night temperature of 25 °C was programmed to increase to 30 °C from 6:00 a.m. to midday gradually. After 2 h at 30 °C, the temperature was gradually reduced, reaching 25 °C at 9:00 p.m. The plants were regularly watered to maintain optimum soil moisture until different stress treatments were started.

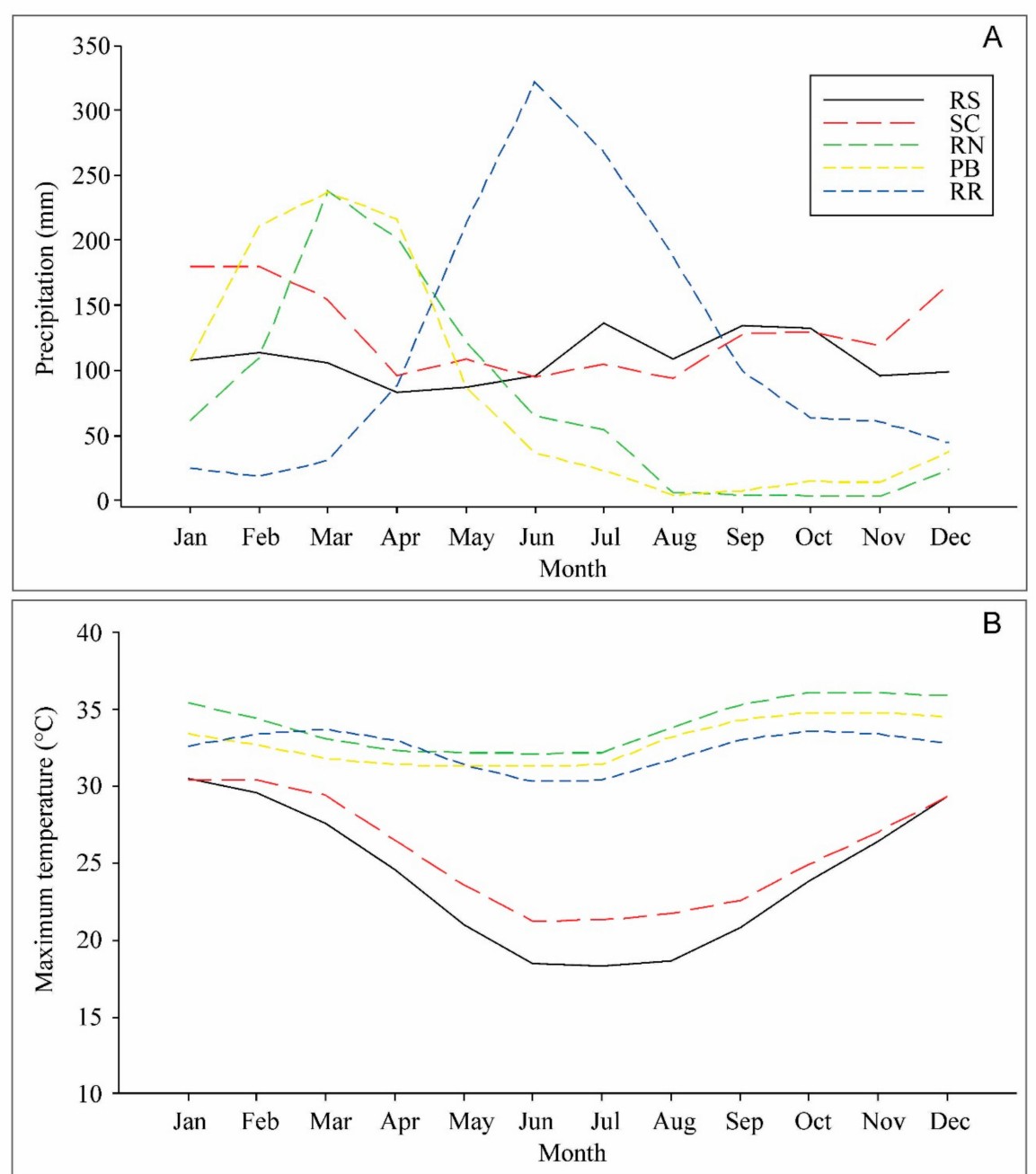

**Figure 1.** Climatic profile in the regions from where weedy rice accessions and cultivated rice were collected. Average rainfall (mm) (**A**); Maximum temperature (°C) (**B**). Source: Adapted from Diniz et al. [50].

For the drought experiment, a preliminary assay was performed to determine the field capacity water volume. The direct gravimetric method was used, using samples for determining the wet weight (after soil was saturated with water) and samples for determining the dry weight (after 48 h in an oven with a temperature between 105–110 °C) [51]. Thirty days after germination (stage $V_3$–$V_4$) [52], the plants were subjected to drought stress by withholding water until the soil moisture dropped to 50% of the field capacity (FC) weight. The plants were kept under stress for ten days. During the drought stress period, pots were weighed daily, at the same time, and the lost water was replenished to maintain 50% FC (Supplementary Figure S2) [53]. The control plants were kept at 100% FC.

Plants submitted to heat stress were kept in the same conditions as the control plants until the stress induction. On the same day when the drought stress treatment was

completed, a group of control plants was moved to a growth chamber set at 42 °C for 2 h [27] to analyze heat stress-responsive genes. The plants were arranged in a completely randomized design with three replicates.

### 2.3. Determination of Photosynthetic Parameters

Photosynthetic parameters were measured immediately after the plants were subjected to 10 d of drought and 2 h of heat stress. The evaluations were performed using a portable photosynthesis meter LI-COR 6400XT (Lincoln, NE, US) at a concentration of 370 $\mu mol \cdot mol^{-1}$ of $CO_2$, the light intensity of 1000 $\mu mol \cdot m^2 \cdot s^{-1}$, and relative humidity between 55 and 60% [54]. The parameters of net photosynthetic rate (A), stomatal conductance ($G_s$), intercellular $CO_2$ concentration ($C_i$), and transpiration rate (E) were analyzed. The instantaneous water use efficiency (WUE) was calculated as the net photosynthetic rate (A) divided by the transpiration rate (E) [55].

The chlorophyll content was measured with a portable chlorophyll meter SPAD-502 (Minolta, Tokyo, Japan) in three replicates in each plant in the central third of the flag leaf. The wavelengths chosen to measure the chlorophyll content were in the red range, wherein the absorbance is high and is not affected by carotenoids, and in the infrared spectrum, wherein the absorbance is extremely low. The transmitted light, which depends on the leaf shade of green, is converted into electrical signals, and the ratio of the intensities of the transmitted light in the two wavelength regions corresponds to a numerical value, called the SPAD (Soil Plant Analysis Development) reading [56].

### 2.4. Morphological Data and Yield Determination

Plant heights were measured 14 d after the stress treatments Plants submitted to either drought or heat temperature stress were kept in the growth chamber, under a 14-h photoperiod and temperature of 25 °C (night) to 30 °C (day) until harvest. The plants were placed in a permanent flood until maturity. Fertilizer was applied as needed, based on the technical recommendations for irrigated rice production in southern Brazil [52]. Tillers were counted when the plants reached the reproductive stage. When the rice grains reached 22% moisture, all panicles were harvested from each plant, and clean seeds were weighed.

### 2.5. Statistical Analysis

The data were submitted to an Analysis of Variance (ANOVA), and means of the control condition were separated by Tukey's test at 5% of probability to verify differences among the genotypes. Both ANOVA and Tukey's test were performed using the Genes software [57]. The drought and heat stress effects in each genotype were verified by calculating the percentage of change compared to the non-treated control (optimal condition of growth), and differences among the genotypes in each stress condition were verified through 95% confidence intervals. Pearson correlation coefficient was calculated using Orange software [58] to identify relationships among the evaluated traits.

### 2.6. Gene Expression Analysis by qRT-PCR

The total RNA was extracted from leaves of rice and WR plants collected after the period of each stress condition using TRIzol Reagent® (Invitrogen™) following the manufacturer's recommendations. The RNA quality and integrity were determined by Nanodrop™ and agarose gel electrophoresis to view RNA integrity and qualitatively assess quantity. RNA samples were treated with DNAse I Amplification Grade (Invitrogen™) to remove genomic DNA. The cDNA synthesis was performed using Reverse Transcription System® (Promega™) from 2 μg of treated RNA using Oligo(dT).

The qRT-PCR was performed following MIQE guidelines (Bustin et al., 2009), using specific oligonucleotides for *OsHSPs* and *OsHSFs* target rice genes and *OsUBC-E2*, *OsUBQ5*, and *OsACT1* as housekeeping genes (Table 2). Validation experiments were performed using four dilutions of cDNA to determine each oligonucleotide's amplification efficiency and specificity. Only oligonucleotides that showed efficiency between 90 and 110% and

with only one peak in the dissociation curve were used. Expression data were subjected to stability analysis using DataAssist™ v3.0 Software (Applied biosystems) from which *OsUBC-E2* and *OsUBQ5* showed score values below 1.0 and were used to normalize the expression data of the *OsHSP* and *OsHSF* target genes.

**Table 2.** Oligonucleotides used for the qRT-PCR assay.

| Gene | Family/Name | Oligonucleotide Sequences (5′→3′) | References |
|------|-------------|-----------------------------------|------------|
| *OsHSP80.2* | *HSP90* | F: CGACGACGAGCAGTATGT<br>R: CCAGATGTTCCTCCCAGT | [59] |
| *OsHSP74.8* | *HSP90* | F: CGAGCAGTTCGAGTACCAGG<br>R: TCAGCCATAGCTTCCCATAC | [60] |
| *OsHSP24.15* | *sHSP* | F: GATCAAGGCGGAGATGAAGAAC<br>R: ACTCGACGTTGACCTGGAAGA | [61] |
| *OsHsfA7* | *HSF* | F: TTCGCCAGCTCAACACCTA<br>R: TCCATCAGCCGTTGTCTT | [38] |
| *OsHsfA2a* | *HSF* | F: TTCGTAGGGTGACGTAATCG<br>R: TCGAAGCCACCGTCCTAG | [44] |
| UBC-E2 | *Ubiquitin-conjugating enzyme E2* | F: CCGTTTGTAGAGCCATAATTGCA<br>R: AGGTTGCCTGAGTCACAGTTAAGTG | [62] |
| UBQ5 | *Ubiquitin 5* | F: ACCACTTCGACCGCCACTACT<br>R: ACGCCTAAGCCTGCTGGTT | [62] |
| ACT11 | *Actin 11* | F: CAGCCACACTGTCCCCATCTA<br>R: AGCAAGGTCGAGACGAAGGA | [62] |

The gene expression assays were performed in BioRad CFX96™ Real-Time PCR Detection Systems, using iQ™ SYBR® Green Supermix (BioRad™) thermocycler. Three biological replicates and three technical replicates were used. The relative expression was calculated using the ΔΔCt method [63]. Gene expression data were analyzed using the MultiExperiment Viewer (MeV) software [64] and presented as a heat map diagram using the control condition of each cultivar/genotype as a baseline to determine RNA levels.

## 3. Results

The interaction effect of cultivars/genotypes and stress factors was significant for tiller number, plant height, photosynthetic rate (A), stomatal conductance ($G_s$), intercellular $CO_2$ concentration ($C_i$), transpiration (E), and water use efficiency (WUE) (Supplementary Table S1). Differences were observed for SPAD and yield across stress treatments and genotypes.

### 3.1. Morphological Traits under Heat and Drought Stress

The weedy rice from Northern Brazil (WR_RR) was taller (76.7 cm) compared to all others when grown under normal conditions (Figure 2A). All rice and weedy rice genotypes were negatively affected by drought, while the WR_RR genotype was most affected by drought stress, showing a 29% reduction in plant height (Figure 2B). Weedy rice from RN (WR_RN) was also significantly stunted by drought, showing a 26% reduction in plant height. All genotypes were negatively affected by heat stress, except for WR_RN, which showed a 5% increase in height.

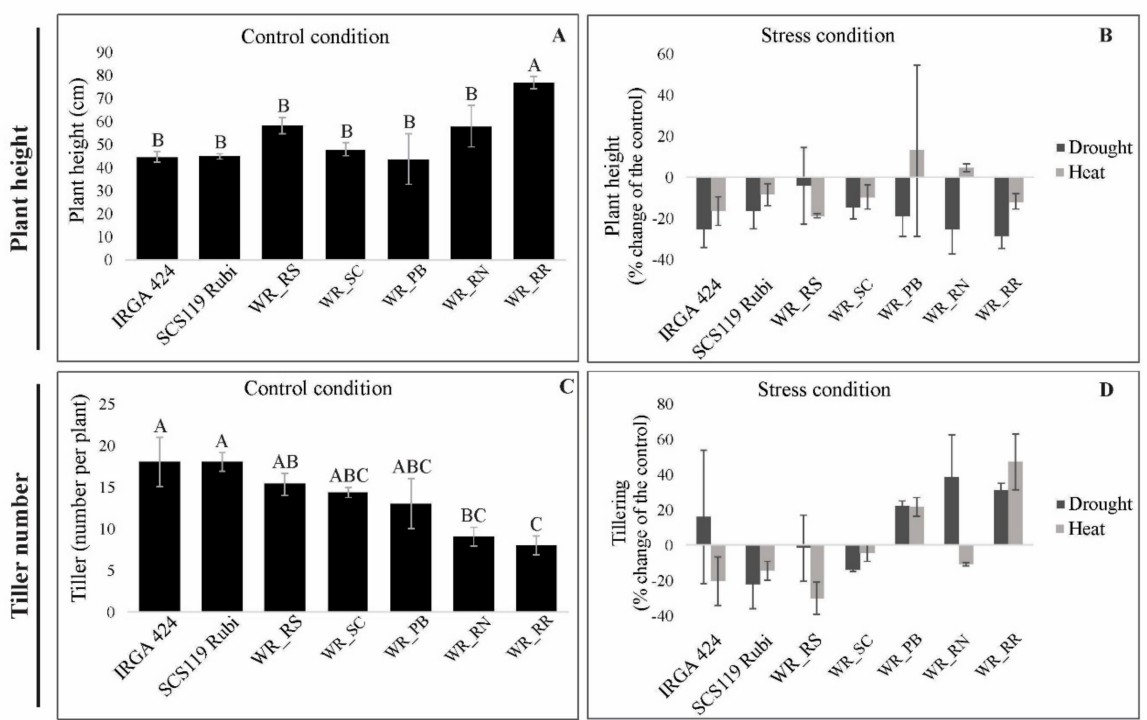

**Figure 2.** Morphological analysis of rice and weedy rice genotypes. Plant height in control (**A**) and in stress condition (**B**); tiller number in control (**C**) and in stress condition (**D**). Means followed by different uppercase letters are different based on Tukey's test ($p < 0.05$). Error bars represent 95% confidence intervals.

The panicle number is a yield component dependent on the tiller number. The highest tiller number was observed for the two rice cultivars: IRGA 424 and SCS119 Rubi (Figure 2C). On the other hand, the WR_RR genotype showed the lowest tiller number among all genotypes. The tiller number of IRGA 424 increased by 15% in drought and decreased 20% in heat stress, similar to WR_RN that showed increasing of 38% in drought and a reduction of 10% in heat stress (Figure 2D). The tiller number of SCS119 Rubi, WR_RS, and WR_SC declined in drought and heat stress conditions. The tiller numbers of WR_PB and WR_RR increased under both stress conditions.

*3.2. Photosynthetic Parameters in Heat and Drought Stress*

Under normal growing conditions, the rice cultivars and weedy rice genotypes did not differ in photosynthesis performance, except for IRGA 424, which had a lower photosynthesis rate (A) than all other genotypes (Table 3). Drought stress reduced A of all analyzed genotypes, with IRGA 424, WR_PB, and SCS119 Rubi being the most affected, showing 59, 63, and 65% drop in A, respectively (Figure 3A). Interestingly, A increased in all genotypes under heat stress with IRGA 424, SCS119 Rubi, and WR_RN showing higher increases in A relative to their non-stressed controls at 19, 24, and 21%, respectively. Stomatal conductance ($G_s$) was not different among the genotypes in optimal condition; however, $G_s$ was reduced under heat stress in all genotypes (Table 3 and Figure 3B). All genotypes showed a reduction in $G_s$ under drought stress, as the stomata close to minimize transpiration and preserve cell turgor. Changes in $G_s$ are associated with the intercellular $CO_2$ concentration ($C_i$). Under optimal conditions, the genotypes did not differ in $C_i$ values (Table 3). Under drought stress, IRGA424, SCS119 Rubi, and WR_PB had increased $C_i$ by 24, 38, and 24%, respectively (Figure 3C). Under heat stress, $C_i$ declined in all genotypes.

**Table 3.** Means of photosynthetic rate (A), stomatal conductance ($G_s$), intercellular $CO_2$ concentration ($C_i$), transpiration rate (E), water use efficiency (WUE), and chlorophyll meter index (SPAD) in rice and weedy rice under controlled conditions. Values within parentheses represent a 95% confidence interval.

| Genotype | A ($\mu$mol $CO_2$ m$^{-2}$ s$^{-1}$) | $G_s$ (mol $H_2O$ m$^{-2}$ s$^{-1}$) | $C_i$ ($\mu$mol mol$^{-1}$) | E (mmol $H_2O$ m$^{-2}$ s$^{-1}$) | WUE (A/E Ratio) | SPAD |
|---|---|---|---|---|---|---|
| IRGA 424 | 18.87 (2.06) B[1] | 0.19 (0.03) A | 272.10 (28.16) A | 3.06 (0.62) A | 6.35 (1.70) A | 40.68 (0.64) A |
| SCS119 Rubi | 19.75 (0.35) AB | 0.16 (0.01) A | 248.57 (12.68) A | 2.58 (0.27) A | 7.69 (0.67) A | 43.52 (1.72) A |
| WR_RS | 20.68 (1.45) AB | 0.19 (0.04) A | 259.44 (39.39) A | 3.15 (0.79) A | 6.85 (2.26) A | 41.49 (2.52) A |
| WR_SC | 20.88 (0.08) AB | 0.20 (0.06) A | 266.60 (38.43) A | 3.39 (1.02) A | 6.51 (2.27) A | 40.91 (0.58) A |
| WR_PB | 22.75 (1.02) AB | 0.24 (0.02) A | 276.77 (11.62) A | 3.94 (0.35) A | 5.80 (0.68) A | 41.80 (2.95) A |
| WR_RN | 24.16 (2.13) AB | 0.22 (0.02) A | 260.60 (19.00) A | 3.65 (0.43) A | 6.68 (1.09) A | 42.96 (1.62) A |
| WR_RR | 26.65 (3.38) A | 0.21 (0.03) A | 244.30 (28.71) A | 3.57 (0.62) A | 7.58 (1.64) A | 39.09 (0.66) A |

[1] Means followed by capital letters, comparing genotypes, differ by Tukey test ($p < 0.05$).

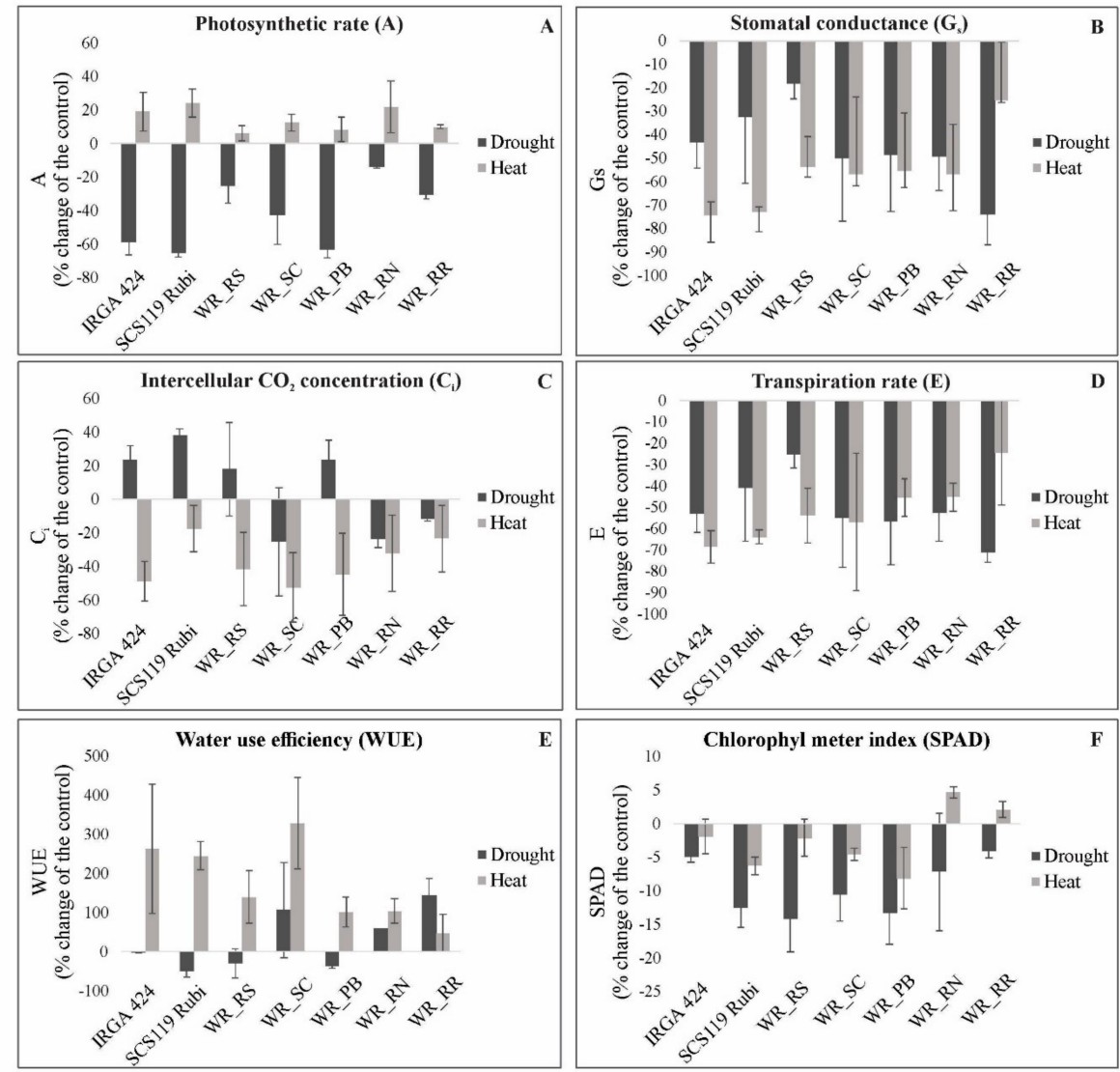

**Figure 3.** Photosynthetic parameters in rice and weedy rice genotypes in drought and heat stress conditions. Photosynthetic rate (A) (**A**); Stomatal conductance ($G_s$) (**B**); Intercellular $CO_2$ concentration ($C_i$) (**C**); Transpiration rate (E) (**D**); Water use efficiency (WUE) (**E**) and Chlorophyll meter index (SPAD) (**F**). Means followed by different uppercase letters are different based on Tukey's test ($p < 0.05$). Error bars represent 95% confidence intervals.

Under drought and heat stress, water loss is modulated by stomatal control of transpiration (E). In normal conditions, the genotypes had similar E rates (Table 3). All genotypes showed a reduction in E in both abiotic stress conditions (Figure 3D). The rice genotypes did not differ in WUE without stress (Table 3). The weedy genotype WR_RR showed higher WUE under drought, increasing by 143% relative to the control plants (Figure 3E). All genotypes showed increased WUE under heat stress, and IRGA 424, SCS119 Rubi, and WR_SC showed higher values at 263, 245, and 328% increase, respectively. Interestingly, WR_RN showed higher WUE both under drought and heat stress conditions with gains at 60 and 104%, respectively.

Drought stress reduced chlorophyll content in all genotypes (Figure 3F). Despite most genotypes showing reduced SPAD values under heat stress, WR_RN and WR_RR showed slightly increase SPAD values at 5 and 2%, respectively.

### 3.3. Grain Yield under Heat and Drought Stress

Under optimal conditions, SCS119 Rubi, WR_RS, WR_PB, WR_RN, and WR_RR yielded better than rice IRGA 424 (Figure 4A). However, WR_SC and WR_RN yield were not affected by heat, while all genotypes were affected by drought stress (Figure 4B). Interestingly, WR_RN was one of the genotypes showing the higher yield in the control condition and was the higher yielder in heat stress, showing good performance in a hot environment.

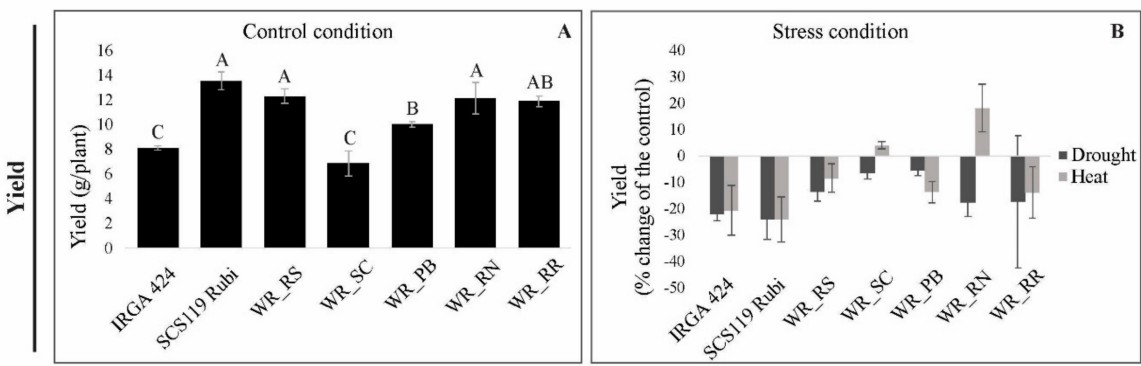

**Figure 4.** Grain yield of rice and weedy rice genotypes in control (**A**) and stress condition (**B**). Means followed by different uppercase letters are different based on Tukey's test ($p < 0.05$). Vertical bars represent 95% confidence intervals.

### 3.4. Gene Expression Analysis under Heat and Drought Conditions

#### 3.4.1. Transcriptional Regulators—HSFs

We analyzed the expression of two stress-responsive *HSFA* genes, *HSFA7* and *HSFA2a*. Drought stress-induced *HSFA7* expression in rice cultivars IRGA 424 and SCS119 Rubi, and WR_RS, showing 3.2-, 4.0- and 2.6-fold, respectively (Figure 5A). On the contrary, *HSFA7* was strongly downregulated (-−4.0 –fold) by drought in WR_SC. Additionally, *HSFA7* was strongly downregulated by heat stress in WR_SC, WR_RS, and WR_PB (−5.3, −2.8, and −1.7 –fold, respectively) and was upregulated only in SCS119 Rubi (3.6 –fold). On the other hand, *HSFA2a* was upregulated in all rice and weedy rice genotypes under heat stress (from 6 to 10 –fold), excepted in WR_RN biotype (−6.0 –fold), but was downregulated or remained the same in most rice and weedy rice genotypes under drought stress (Figure 5A).

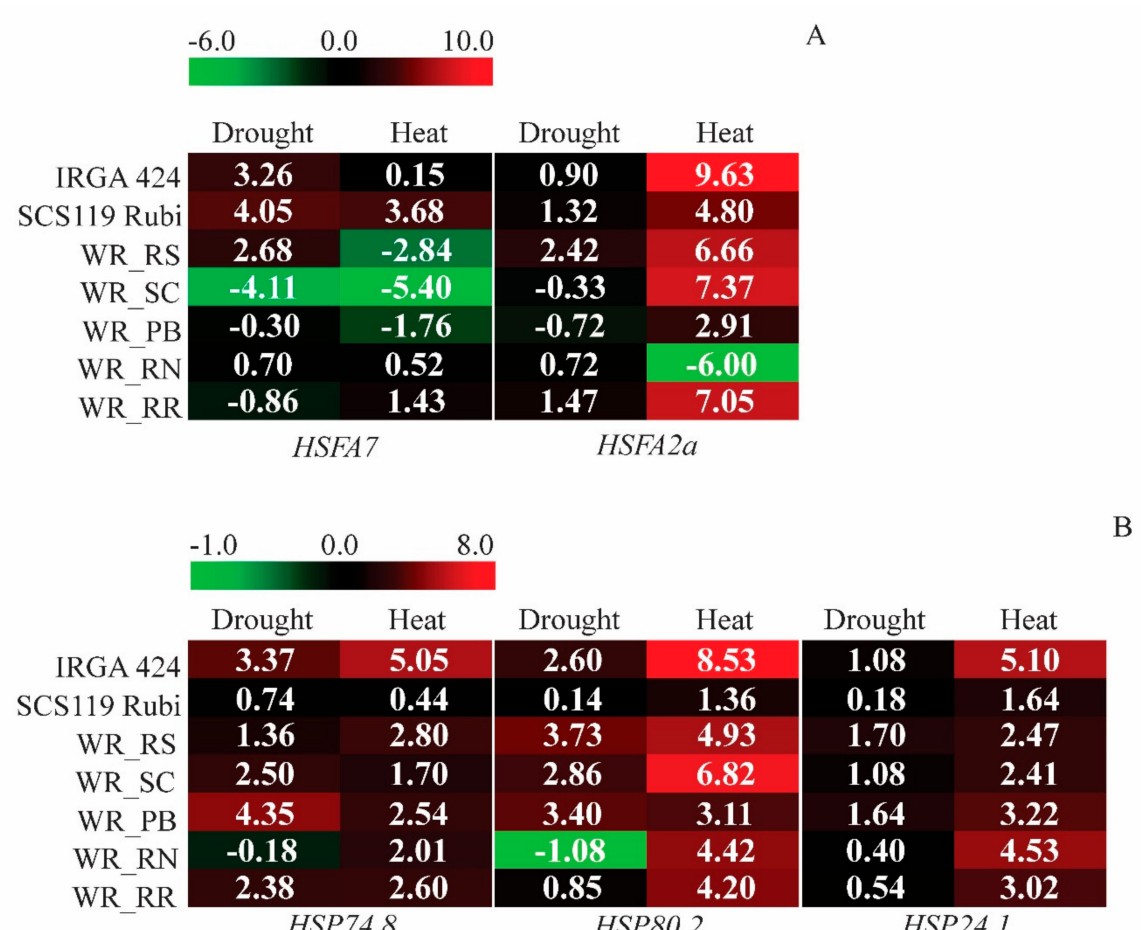

**Figure 5.** Relative mRNA abundance in rice and weedy rice leaves. Relative expression of *HSF* genes (**A**); Relative expression of *HSP* genes (**B**). mRNA abundance is represented in Log$_2$ fold-change using the Multi Experiment Viewer (TIGR MeV) software [64]. The color scale above the heat map shows the expression level; red = high transcript abundance, green = low transcript abundance.

3.4.2. Stress-Responsive Genes—HSPs

Considering the critical role of HSPs in stress signaling and mainly of HSP90 proteins, we analyzed the expression of two HSP90 genes, *HSP74.8* and *HSP80.2* [59]. In general, *HSP74.8* showed transcript accumulation changes in heat and drought stress for most genotypes studied (Figure 5B). Specifically, *HSP74.8* was upregulated in both stress conditions in all rice cultivars and weed rice genotypes (from 1.6 to 5.0 –fold), except WR_RN weedy rice, which was unaffected, under drought stress. *HSP80.2* was also upregulated in all genotypes under heat stress (from 3.11– to 8.0 –fold) (Figure 5B). The expression of *HSP80.2* was not affected by drought in SCS119 Rubi (1.3 –fold) and WR_RN (−1.01 –fold) and WR_RR (0.8 –fold).

Small heat shock proteins (sHSPs) belong to a ubiquitous family of chaperones that act independently of ATP and are the first defense line in the cell against stress [65]. Thus, knowing the vital role of sHSPs in stress responses, we studied the expression of the *sHSP24.1* [59]. *sHSP24.1* was expressed in both stress conditions; however, its expression was only barely induced by drought (1.68 -fold) and significantly induced by heat stress (up to 5.1 -fold) (Figure 5B).

### *3.5. Relating Physiological and Molecular Responses to Phenotypic Responses*

### 3.5.1. Plant Height

Under heat stress, $G_s$ and E were strongly positively correlated with plant height (Figure 6A). Thus, plants with high $G_s$ and E are supposed to be tall. As $G_s$ values dropped under heat stress, so did plant heights (Figures 2B and 3B,D). The $C_i$ was also positively correlated with plant height, but this relationship was weak. In this sense, tall plants did not necessarily always have high $C_i$ values under heat stress. The expression of *HSF2a* was strongly and positively correlated with plant height under drought stress and weakly, negatively correlated with height under heat stress (Figure 6A). Plants under heat stress were shorter than those under drought stress. Despite the correlation values, the expression of *HSF2a* was not affected by drought stress, while plants were smaller in this condition compared to those under heat stress (Figures 2B and 5A).

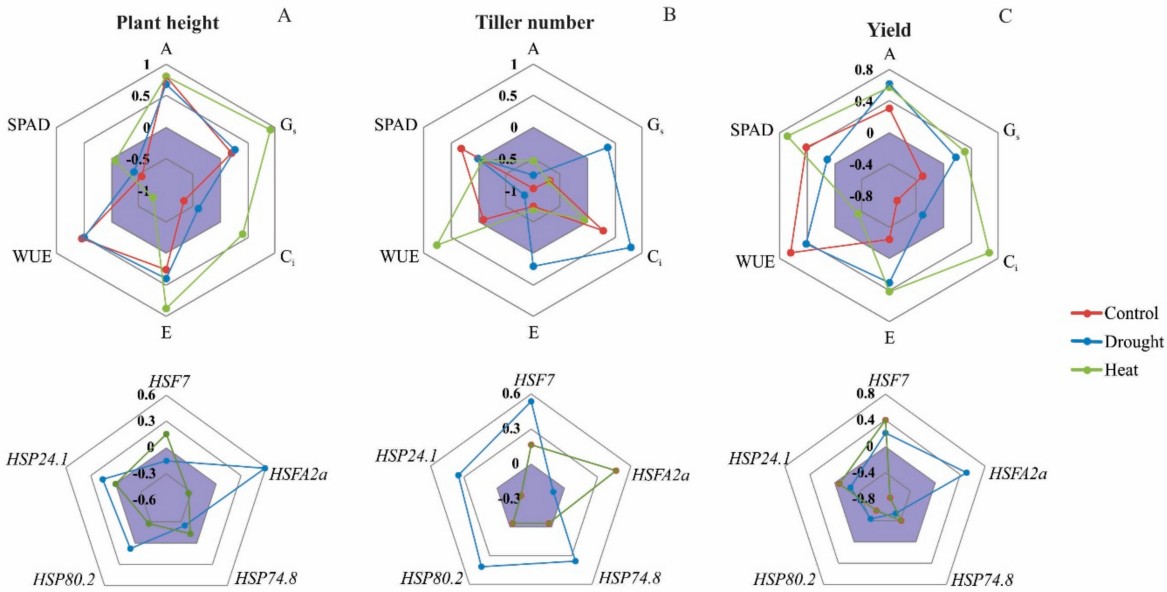

**Figure 6.** Graphic representation of Pearson correlations between photosynthesis-related variables and *HSFs* and *HSPs* gene expression in rice cultivars and weedy rice genotypes under normal growing conditions, drought stress, and heat stress. Plant height correlations (**A**); Tiller number correlations (**B**); Yield correlations (**C**). Photosynthesis rate (A); Stomatal conductance ($G_s$); Intercellular $CO_2$ concentration ($C_i$); Transpiration rate (E); Water use efficiency (WUE); Chlorophyll meter index (SPAD). The shaded area at the center represents a negative correlation.

### 3.5.2. Tiller Number

Under drought stress, WUE was strongly negatively correlated with tiller number (Figure 6B). Except for WUE that was significantly positively correlated with heat stress, all the other photosynthetic parameters were strongly negatively correlated with heat stress (Figure 6B). WR_PB and WR_RR genotypes showed increased tiller number (21 and 47%, respectively) and increased WUE (Figures 2D and 3E); hence are most likely heat-tolerant plants. Considering that almost all genotypes had reduced tiller numbers and increased WUE under heat stress, WUE is not a good standard to identify heat-tolerant plants that would produce more tillers under high temperature (Figures 2D and 3E). WUE also cannot be used to discriminate drought-tolerant plants. IRGA 424, WR_RS, and WR_SC genotypes displayed a negative correlation between WUE and tiller number. A strong positive correlation was observed between $C_i$ and tiller number under drought stress, while a weak positive correlation was observed under heat stress. As observed for WUE, $C_i$ is not a suitable standard for selecting drought-tolerant plants since the increase of $C_i$ and an increase in tiller number were detected only in IRGA 424 and WR_PB genotypes.

### 3.5.3. Yield

Under drought stress, WUE was positively correlated with yield, while under heat stress, WUE and yield were negatively correlated (Figure 6C). In general, under drought stress, genotypes that showed a reduction in WUE also had reduced yield (IRGA 424 and SCS119 Rubi); however, those with increased WUE (WR_SC, WR_RN, and WR_RR) also had yield loss under drought (Figures 3E and 4B). Under heat stress, most genotypes showed a reduction in yield and an increase in WUE, reflecting the negative correlation of these two variables under this condition. The WR_RN weedy rice was an exception, displaying an increased yield and increased WUE. SPAD and $C_i$ values were positively correlated with yield under heat stress. In almost all genotypes, SPAD decreased under heat stress while yield also decreased (Figures 3F and 4B). The reduction in $C_i$ in all genotypes under heat stress also correlated with reduced yields (Figures 3C and 4B).

The expression of the TF *HSF2a* was positively correlated with yield under drought stress and strongly negatively correlated with yield under heat stress (Figure 6C). *HSF2a* did not change their expression in drought conditions, while yield was reduced compared to the control condition (Figures 4B and 5A). Under heat stress, *HSF2a* was upregulated in almost all genotypes, and yield was reduced. The expression of *HSP80.2* was negatively correlated with yield under heat stress and was upregulated in all genotypes.

## 4. Discussion

### 4.1. Physiological Parameters Indicate Genotypes with Suitable Performance under Drought and Heat Stresses

Plants growing under distinct environmental stress conditions often undergo morphological and physiological adjustments to deal with these challenges [66]. Heat and drought stress are major abiotic stresses affecting crops. When soil moisture and atmospheric humidity are low, the water uptake is low while evapotranspiration is high, creating drought stress in the plant. On the other hand, heat stress increases air and soil temperature beyond a threshold level [67]. Heat stress can disturb carbohydrate metabolism, affecting plant growth and development, pollen viability, and ovule fertilization leading to spikelet sterility and loss of grain yield in rice [68]. Drought stress also causes morphological effects in plant growth since cell elongation is inhibited by interruption of water flow (loss of turgor), yield, membrane integrity, pigment content, and photosynthesis [69]. Stunting is one of the most critical morphological effects of drought stress.

Plant height of all genotypes cultivated rice and weedy rice were negatively affected by drought stress. Reduction in plant growth was also reported previously in rice cultivars submitted to drought stress during the vegetative stage [70]. Heat stress also negatively affects plant height; however, WR_RN increased its height in heat stress conditions. One of the heat stress effects in rice's vegetative stage is the decrease in the $CO_2$ assimilation rate by affecting the carbon metabolism and photochemical reactions in the chloroplast [71]. It may explain the negative effect on plant growth detected here. Physiological parameters such as relative water content, leaf water potential, stomatal conductance, transpiration rate, and leaf temperature are modulated in limited water- environments such as drought and heat [69,71,72]. WR_RN does not show increased $C_i$ and $G_s$ but showed the highest A and SPAD in heat stress conditions, explaining the increased height in this condition. Sustainment of leaf gas exchange and $CO_2$ assimilation rates is directly correlated with heat tolerance [71].

There are differences in A among the genotypes, which was not expected since the rice cultivars and weedy rice belong to the same species and are C3 plants. WR_RN shows the highest A in the control condition. However, the activation of 1,5-bisphosphate carboxylase/oxygenase (Rubisco) declines under heat stress since heat inhibits the activity of Rubisco activase. The result is the reduction of A under heat stress [73,74]. Thus, the increase in A identified here for all genotypes, mainly for WR_RN and the rice cultivars, suggests that these genotypes have suitable performance under heat stress. The amount of

Rubisco is affected by heat and drought stress, affecting Rubisco activation in rice [75]. It may explain the reduced A found here in all genotypes under drought stress.

Abiotic stresses such as drought also inhibit chlorophyll synthesis [69]. The decline in chlorophyll content is associated with chloroplast damage caused by reactive oxygen species (ROS) [76]. Our data corroborated this, showing that drought stress reduced chlorophyll content in all genotypes. WR_RN and WR_RR show a slight increase SPAD under heat stress. SPAD readings in rice leaves are significantly correlated with the chlorophyll content, Rubisco content, and gross photosynthetic rate [77]. This indicates that SPAD readings are an indirect indication of the photosynthetic capacity of rice leaves. Thus, it can be suggested that WR_RN and WR_RR genotypes are good candidates to improve rice's photosynthetic capacity under hot environments. Additionally, as temperature rises, stomata generally close to reduce transpiration loss, reducing the diffusion of atmospheric $CO_2$ into the leaf and consequently reducing the intercellular ($CO_2$) in the leaf mesophyll cells [78]. In drought conditions, ABA accumulates within the leaves leading to stomatal closure to reduce transpiration water loss. The decline in $G_s$ associated with the higher ABA content lowers the availability of $CO_2$ at the site of carboxylation, resulting in reduced A and growth [74]. Our data demonstrate this since all the stressful conditions cause a reduction in $G_s$. The $C_i$ was increased in IRGA424, SCS119 Rubi, and WR_PB genotypes in drought stress conditions, demonstrating a suitable performance of these genotypes under this condition.

Also, under drought and heat stress conditions, water loss is modulated by stomatal control of E. Reducing E when water is limiting is an effective means of avoiding or mitigating the effect of drought. Drought avoidance is a tolerance mechanism that involves reducing water loss (conserving water) mainly by stomatal control of transpiration [72]. In this sense, it seems that the weedy rice genotype WR_RR harbors a drought avoidance mechanism to cope with drought stress.

The rising temperature and [$CO_2$] and variations in humidity and precipitation influence water use by plants. This dynamic directly affects WUE, which is the amount of carbon assimilated as biomass for every unit of water taken up and is calculated by dividing A by E [55]. Improvement of WUE is necessary for the sustainability of rice production in water-limited areas [79]. In leaves, water use is regulated by $G_s$. All analyzed genotypes show increased WUE under heat, but only WR_SC, WR_RN, and WR_RR show increased WUE in drought conditions. These genotypes are good candidates as a source of variation to improve rice WUE.

*4.2. Relationship between Photosynthetic Parameters and Grain Yield Reveals Genotypes with Suitable Performance under Drought and Heat Stresses*

By affecting many important morphological traits and photosynthetic parameters, abiotic stresses negatively affect crop yield. One of the main effects is the damage incurred in the reproductive components such as male sterility, flower abortion, and fruit drop. These effects may reduce yield volume and quality [80]. Heat and drought stress are significant challenges to sustain rice yields. With each 1 °C increase during the nighttime, rice grain yield decreases by 10% [81]. The accumulation of essential stress-responsive amino acids and protective metabolites and the sucrose supply regulation to avoid carbohydrate starvation are adaptations that prevent reproductive failure in rice under heat and drought stress [82]. The rice TF HIGHER YIELD RICE (HYR) is reported as a master regulator directly activating photosynthesis genes, regulating TF cascades and other downstream genes involved in photosynthetic carbon metabolism, and enabling yield stability under drought and heat stress [8]. This finding demonstrated that coordinated regulation of photosynthesis leads to an increase in rice yield. Generally, water deficit reduces yield, which could be the ultimate effect of reduced $CO_2$ diffusion into the leaf due to stomatal closure to minimize water loss via transpiration [72].

WR_SC weedy rice was the least affected in yield in both stress treatments. All genotypes suffered some yield loss under drought or heat stress, except for WR_RN weedy rice, which produced 18% more grains under heat stress relative to control plants.

Interestingly, the WR_RN photosynthetic responses under heat stress show 22% higher A, 57% less $G_s$, 31% lower $C_i$, 45% lower E, 104% higher WUE, and 5% higher SPAD value relative to the control plants. Altogether, this shows that WR_RN has a better performance in hot environments. These data suggest that WR_RN may harbor a C3–C4 intermediate pathway. Plants with the C3–C4 intermediate photosynthesis are typically from hot habitats (as in WR_RN). Such plants exhibit an adaptive trait of re-assimilating photorespired $CO_2$ at high temperatures [83]. The photorespired $CO_2$ is re-assimilated through the differential partitioning of organelles involved in photorespiration between mesophyll cells and bundle-sheath cells [83]. One of the C4 plants' main traits is the presence of chloroplasts in bundle-sheath cells and the increased size of these cells, helping plants fix $CO_2$ [84]. Generally, rice plants do not have chloroplasts in bundle sheath cells. However, rice genotypes showing chloroplasts in bundle-sheath cells were identified, and it is hypothesized that this genotype will perform better in hot environments [85]. Ultrastructural analysis of WR_RN bundle-sheath cells can help identify the mechanism by which this weedy genotype yields better in hot conditions.

### 4.3. Modulation of HSFs and HSPs under Drought and Heat Stresses

Besides the morphological adaptations discussed above, plants also have evolved sophisticated cellular, and molecular systems wherein heat shock proteins (HSPs) are produced under stress [80]. Sensors and receptors in the cell membrane signal the onset of heat or drought stress, activating secondary messengers, especially $Ca^{2+}$ ions, which activate signaling pathways to transduce downstream signals. MAPKs (mitogen-activated protein kinases) and CDPK (calcium-dependent protein kinases) activate TFs (from many different families, including HSFs). These TFs transcriptionally regulate stress-related proteins, which act as chaperones (such as HSPs), effecting heat, and drought tolerance [86]. Under stress, HSFs bind to cis-acting heat stress elements (HSEs) in promoters of stress-inducible genes like *HSPs*, performing a central role in plant stress tolerance [87]. HSPs are proteins that facilitate the proper folding of newly synthesized proteins or prevent protein folding during stress to preserve cell function [80]. Thus, known plant molecular responses under stress, combined with morphological and physiological profiles, help understand stress tolerance dynamics and potentially harness such mechanisms to improve crop stress tolerance and productivity.

Plant HSFs are classified into three major classes, A, B, and C, based on their protein structure [40]. Many of the HSFs from class A was reported to be involved in plant response to a wide range of abiotic stresses, such as osmotic and oxidative stresses in rice [38,44] and other plant species [88].

Among the rice *HSF* genes, *HSFA2a* is reported as one of the most responsive in the first 24 h of exposure to abiotic stresses, including water deficit, heat, salinity, and cold [44]. This early response implies that *HSFA2a* is involved in plants' primary response to different types of abiotic stress. Here we identified that *HSFA2a* was strongly upregulated by heat and downregulated or non-activated by drought stress. However, *HSFA2a* expression was negatively correlated with yield. Interestingly, *HSF2a* was down-regulated exclusively in WR_RN, the only genotype able to increase yield under heat stress. These results suggest that the upregulation of *HSF2a* is associated with decreased yield, while its downregulation is associated with increased yield.

*HSFA7* is a slow-response gene [44]. *HSFA7* contributes to drought stress tolerance in rice by inducing HSP expression [38], which may be associated with adaptation to stress. Rice cultivars and WR_RS show strong *HSFA7* upregulation under drought, but *HSFA7* is downregulated or non-activated in heat stress. Interestingly, *HSFA7* was found in a robust co-expression and *cis*-regulatory elements analysis as highly responsive to water deficit. In this analysis, co-expressed genes encode stress-related functional proteins [89]. The target genes are ABREs (abscisic acid-responsive elements), indicating that these genes are probably involved in ABA-dependent responses to water deficit. ABA plays an important role in plant response to osmotic stress, and its accumulation leads to stomata's closing

to curb water loss. ABA also modulates the phosphorylation status of TFs and other regulatory proteins involved in stress response [69,90]. The involvement of *HSF* genes in plant response to drought has been documented in different species, including *Sorghum bicolor* L. [91], potato (*Solanum tuberosum* L.) [92], and Arabidopsis [93]. Additionally, in response to heat stress in *S. bicolor* [91], potato [92], radish (*Raphanus sativus* L.) [94], and *Vitis vinifera* [95] mainly *HSFA2* and *HSFA7* genes [96].

Heat shock proteins (HSPs) are activated by HSFs transcription factors and act as molecular chaperones, assisting in protein folding during cellular metabolism, stabilizing membrane proteins, and facilitating protein refolding during stress [86]. Some HSPs act on three-dimensional protein folding of newly formed proteins and proteins damaged by stress within cells [86]. Rice HSPs are classified into sHSP, HSP60, HSP70, HSP90 and HSP100 families with 29, 20, 26, 9 and 5 genes each, respectively [34]. Each HSP family presents a specific mechanism. HSP90 proteins are the most abundant class of HSPs, representing ~1% of the total protein in cells and are found in the cytoplasm [22]. HSP90 contains two highly conserved domains, the ATP-binding domain and a glutamic acid-rich binding domain [97]. All HSP90 proteins present a TPR-binding site for interaction with TPR domain-containing proteins [98]. HSP90 proteins are responsible for denatured protein folding, newly synthesized protein folding, and regulating many cell signaling molecules [99].

Under heat stress, HSP90 proteins can be induced up to 6% of the total protein [100]. Our research also reveals that heat stress-induced *HSP74.8* up to 5-fold in IRGA 424 and up to 2.7 -fold in WR_RS weedy rice. Likewise, *HSP74.8* and *HSP80.2* induction by heat stress in rice has been documented within 3 h of stress implementation [59]. *HSP74.8* was upregulated in rice leaves and stems, implying a possible role in maintaining leaf functions such as respiration and photosynthesis. In Arabidopsis, under normal conditions, the ROF1 protein interacts with HSP90 in the cytoplasm by its binding in the TPR domain [98]. However, under heat stress, ROF1-HSP90.1 complexes with HSFA2 and migrates to the nucleus. There, it regulates the transcription of sHSPs to increase heat tolerance [101]. In agreement, our study also shows transcript accumulation of *HSFA2a* up to 9.6 -fold under heat stress, together with HSP90s *HSP74.8* and *HSP80*.

The expression of *HSP80.2*, coding for HSP90, was negatively correlated, which yield under heat stress. HSP90 facilitates the folding of newly synthesized and denatured proteins and regulates many cell signaling molecules [99]. Because of this critical role, the upregulation of *HSP80.2* under heat stress in almost all genotypes identified here is not surprising. It is a strong indicator of heat stress, which is generally reflected in yield loss. In other words, it helps the plant survive the stress but not mitigate yield loss. Because of its global induction by heat stress, *HSP80.2* cannot be used as a marker for heat-tolerant, yield-stable rice lines.

sHSPs belong to a ubiquitous family of chaperones that act independently of ATP and are the first defense line in the cell against stress when the 3-D structure of proteins begins to unravel [65]. In the present study, the cultivar IRGA 424 and WR_PB, WR_RN, and WR_RR show higher than 3-fold *sHSP24.1* transcript accumulation under heat stress. In a similar study, *sHSP24.1* was also upregulated in rice plants submitted to heat stress (42 °C) for 3 h [61]. The role of *sHSP24.1* in thermotolerance was observed in rice spikelets at the meiosis stage of pollen. In this case, *sHSP24.1* mediates sucrose metabolism under heat stress. The ABA content increases following the *sHSP24.1* induction [68]. It turns out that ABA performs a dual role; it can facilitate the function of *sHSP24.1* and also inhibit it [64]. What triggers positive or negative regulation by ABA is unknown, but this duality may explain the lack of response of *sHSP24.1* expression to drought stress in SCS119 Rubi, WR_RN, and WR_RR genotypes. Additionally, it is interesting to note the *sHSP24.1* upregulation in the WR_RN genotype. Considering this gene's role in thermotolerance in rice spikelets, *sHSP24.1* may be associated with the increased WR_RN yield under heat stress.

The rice sHSP24 is a mitochondria-localized HSP [102]. Mitochondria localized sHSPs have been shown to protect the electron transport chain in mitochondria from abiotic stress damage [103].

### 4.4. Modeling Rice and Weedy Rice Responses to Drought and Heat Stresses

Based on the phenotypic response, photosynthetic parameters, and gene expression profiles of rice and weedy rice genotypes under drought and heat stress, we propose a model of expected responses for a plant to perform well in these stress conditions (Figure 7). In drought stress conditions, it is suggested that to maintain or increase tiller number, an increase in transcript accumulation of the *HSFA7* gene is essential since these traits are positively correlated. Related to yield in drought stress conditions, increase WUE and *HSFA2a* transcriptional activation is suggested to maintain or increase yield in this stress since they are positively correlated with yield.

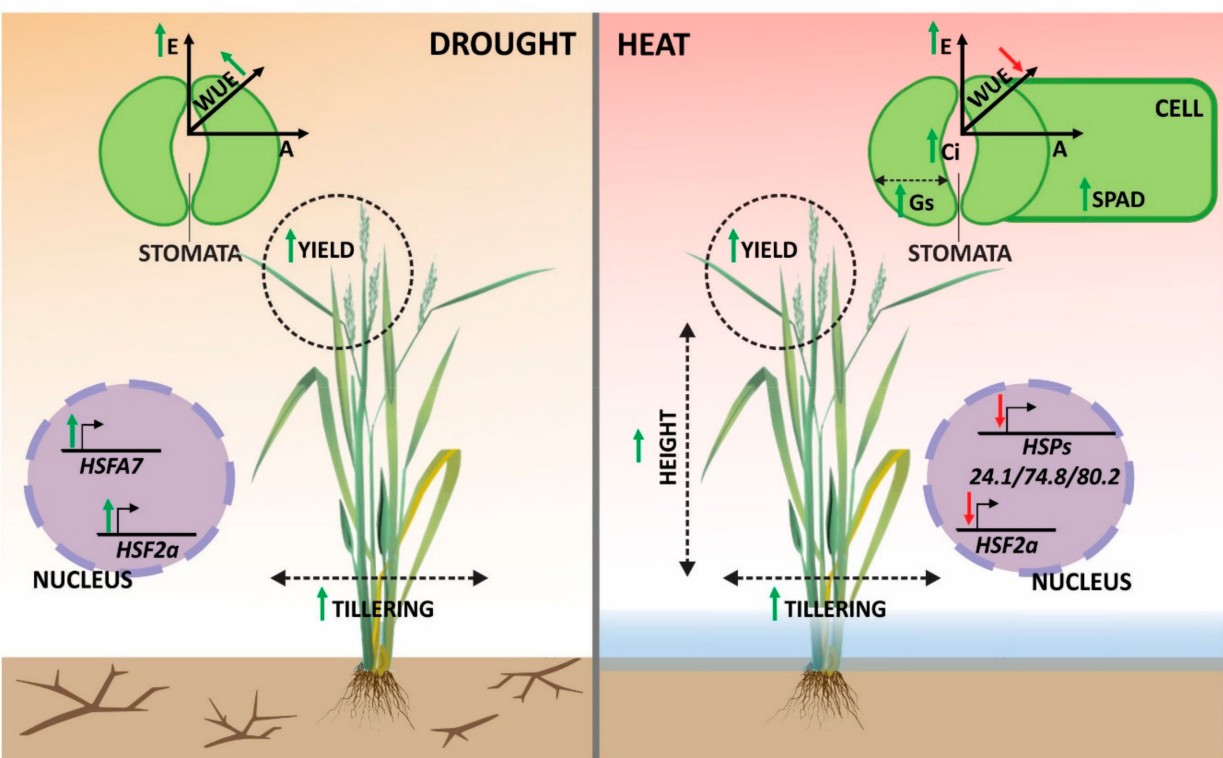

**Figure 7.** Proposed model of morphology, photosynthetic, and molecular responses for suitable performance in drought and heat stress conditions for rice and weedy rice plants. Red arrows mean decrease, and green arrows mean increase.

Under heat stress, photosynthetic parameters $G_s$ and E were positively correlated with plant height, suggesting that increases in these parameters may help to maintain or increase plant height under high temperatures. Tiller number was negatively correlated with the expression of *HSPs* in heat stress, suggesting that maintaining or increasing tiller number is important a non-activation or repression of *HSPs* expression. Finally, the yield was negatively correlated with WUE and *HSFA2a* transcript accumulation but positively correlated with SPAD and $C_i$, suggesting that to maintain or increase yield under heat stress, it is necessary to perturb the E/A ratio, non-activated or repress *HSFA2a*, increase SPAD and $C_i$. Although yield was negatively correlated with WUE in heat stress conditions, it is indispensable to search for genotypes to deal efficiently with the water resource. All genotypes could increase WUE in response to heat stress, and IRGA 424, SCS119 Rubi, and SC genotypes show the best WUE performance. Overall, despite high-temperature occurrence at lower latitudes such as PB, RN, and RR, weedy rice genotypes from higher latitudes RS and SC also presented a suitable performance under heat stress conditions.

## 5. Conclusions

Drought stress for ten days and heat stress at 42 °C for 2 h are suitable for discriminating stress-tolerant genotypes. The yield is reduced by drought in all genotypes, but WR_PB produces more tillers than the control, and the yield is less negatively affected by drought.

Despite the high-temperature occurrence at lower latitudes (PB, RN, and RR), weedy rice genotypes from higher latitudes RS and SC also have good performance under heat stress. WR_RN yielder better under heat stress and has indications of possibly possessing a C3–C4 metabolism. Upregulation of the *sHSP24.1* gene in WR_RN can be linked to increased yield in this condition.

In general, weedy rice genotypes perform better than rice cultivars in stress conditions, showing potential utility for sourcing genes to improve rice crop tolerance to drought or heat stresses.

**Supplementary Materials:** The following are available online at https://www.mdpi.com/2077-0472/11/1/9/s1, Table S1: Summary of analysis of variance of rice and weedy rice under heat and drought stress, Figure S1: Distribution of sites where weedy rice biotypes used in the study were collected. Blue: Dom Pedrito-RS; Orange: Gaspar-SC; Red: São José do Rio do Peixe-PB; Yellow: Apodi-RN; and Purple: Bonfim-RR. Source: Google Maps, 2017, Figure S2: Drought stress induction. A: Rice and weedy rice plants under drought stress; B: Daily water replacement to 50% of the field capacity.

**Author Contributions:** Conceptualization, N.R.-B., J.A.N., L.A.d.A., and L.B.P.; formal analysis, L.B.P. and V.E.V.; investigation, L.B.P., C.d.O.; data curation, L.B.P. and V.E.V.; writing—original draft preparation, L.B.P., N.R.-B., and V.E.V.; writing—review and editing, N.R.-B., L.A.d.A., F.P.L., V.E.V., J.A.N., L.B.P. with approval from all authors.; supervision, J.A.N., F.P.L., L.A.d.A., and N.R.-B.; project administration, J.A.N.; funding acquisition, L.A.d.A., J.A.N., and N.R.-B. All authors have read and agreed to the published version of the manuscript.

**Funding:** This research received funding from: Coordenação de Aperfeiçoamento de Pessoal de Nível Superior-Brasil (CAPES)-Finance Code 001; BASF Corporation grant to N.R.-B.; the University of Arkansas (Hatch Project ARK02223 and ARK02606), Fayetteville, NC, USA; the research fellowship of L.A.d.A. by Conselho Nacional de Desenvolvimento Científico e Tecnológico (CNPq—Proc.N. 310538/2015–7), the student sandwich doctoral fellowship of L.B.P.—Ciência Sem Fronteiras/Public Call number 401381/2014-5 mode MEC/MCTI/CAPES/CNPQ/FAPS—Visiting Researcher fellowship—PVE 2014, and for the sandwich doctoral fellowship Proc.N. 207333/2015-7 CNPq and by the research fellowship of V.E.V. by Coordenação de Aperfeiçoamento de Pessoal de Nível Superior-Brasil (CAPES)/ Programa Nacional de Pós Doutorado (PNPD).

**Institutional Review Board Statement:** Not applicable.

**Informed Consent Statement:** Not applicable.

**Acknowledgments:** The authors thank Gabriela Freitas for her assistance in initiating the gene expression assay by qRT-PCR and the University of Arkansas and Federal University of Pelotas employees for their help.

**Conflicts of Interest:** The authors declare no conflict of interest.

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
