# Peer review of "Molecular and Physiological Responses of Rice and Weedy Rice to Heat and Drought Stressâ€"

_agriculture, doi:10.3390/agriculture11010009_

Round 1
Reviewer 1 Report
This manuscript presents a very good comparative analysis of drought and heat stress responses of a variety of rice cultivars, which is very valuable to inform future practical decisions for cultivar/variety selection and breeding efforts.
I only have a few comments:
- in the abstract a the SC119Rubi is once also presented as SC199Rubi, needs to be corrected.
- there is a reference (which I assume was added in after all references were already numbered by the authors) in lane 134: de Freitas et al., 2016. This reference is not included in the list of references. Please include. That will change the numbering of all subsequent references on the list and the authors will have to change those numbers throughout the entire text.
- The information for control values (in panels A) for figures 2 through 5 would best be presented all in one table rather than in separate panels for those figures. The figures should focus on the % changes the authors detected in the treatments.
Author Response
Cover letter
Dear Reviewer 1,
Your comments were important to improve and ensure the quality of the manuscript. Below we list the reviewer’s comments and our responses.
Reviewer 1: in the abstract a the SC119Rubi is once also presented as SC199Rubi, needs to be corrected.
Authors: The name of the rice cultivar SCS119 Rubi was corrected in the abstract.
Reviewer 1: There is a reference (which I assume was added in after all references were already numbered by the authors) in lane 134: de Freitas et al., 2016. This reference is not included in the list of references. Please include. That will change the numbering of all subsequent references on the list and the authors will have to change those numbers throughout the entire text.
Authors: We included in reference list and formatted de Freitas et al., 2016 according the Agriculture requirements.
Reviewer 1: The information for control values (in panels A) for figures 2 through 5 would best be presented all in one table rather than in separate panels for those figures. The figures should focus on the % changes the authors detected in the treatments.
Authors: We created a table (table 3) to present the information for control values and the figure now focus only on the % changes.

Reviewer 2 Report
The manuscript is interesting and addresses the problem of rice resistance to drought and high temperature and the search for resistance traits among wild genotypes. An interesting result of the work is the modeling of the response of rice to both of these stresses. The work, however, requires many corrections. Overall, the work is wordy. It contains a lot of basic and handbook information, not always matching the studied problem.
Abstract:
There should be an explanation of how many genotypes have been studied: how many cultivars and how many wild forms. A detailed description of the response of individual genotypes, which does not tell the reader much, I would replace with the description of the model presented at the end of the paper.
Introduction:
The LEA protein abbreviation must be explained
Material and methods:
Point 2.4. It is vaguely described. Which plants were transferred to the greenhouse? After what duration of stress? Were they already grown in the greenhouse under optimal thermal and water conditions? Can this period be called ‘recovery’?
Results and Discussion
In Fig. 1. The control bars should be lighter to see the standard deviation. Fig. 1 B and D need to be clearer. They are hardly legible now.
Discussion is the weakest side of the job. The authors do not discuss with the results of other scientists, but quote a number of redundant information. Why they describe the course of photosynthesis? Why compare C3 plants with C4 plants when rice is a C3 plant? Why write about abscisic acid, if its level has not been studied by the authors? The whole fragment about ROS is unnecessary. You can mention these compounds, but they don't need to take up whole paragraphs. Figure 8 with description should be transferred to methods. The most interesting model of rice plants' response to the stresses studied (Fig. 9) should be much more precisely described and explained.
Final conclusion
The work must be rewritten, shortened, the results should be described separately from the discussion. Thanks to this, the work will be more transparent and legible. After the suggested corrections it can be resubmitted.
Author Response
Cover letter
Dear Reviewer 2,
Your comments were important to improve and ensure the quality of the manuscript. Below we list the reviewer’s comments and our responses.
Reviewer 2: The manuscript is interesting and addresses the problem of rice resistance to drought and high temperature and the search for resistance traits among wild genotypes. An interesting result of the work is the modeling of the response of rice to both of these stresses. The work, however, requires many corrections. Overall, the work is wordy. It contains a lot of basic and handbook information, not always matching the studied problem.
Authors: Based in your important comments, we improved the text to insure a better understanding of the studies problem.
Reviewer 2: Abstract: There should be an explanation of how many genotypes have been studied: how many cultivars and how many wild forms. A detailed description of the response of individual genotypes, which does not tell the reader much, I would replace with the description of the model presented at the end of the paper.
Authors: We inserted the information related the number of analyzed genotypes and change the text with the description of the concluding model.
Reviewer 2: Introduction: The LEA protein abbreviation must be explained.
Authors: We inserted the full name of LEA protein in the introduction section.
Reviewer 2: Material and methods: Point 2.4. It is vaguely described. Which plants were transferred to the greenhouse? After what duration of stress? Were they already grown in the greenhouse under optimal thermal and water conditions? Can this period be called ‘recovery’?
Authors: We improved the text with a more precisely description. All pants were kept in a growth chamber; they were not transferred to greenhouse. The duration of the stress is in the text at point 2.2. Plants remained in a growth chamber from germination to stress application. We can call this period as recovery.
Reviewer 2: Results and Discussion: In Fig. 1. The control bars should be lighter to see the standard deviation. Fig. 1 B and D need to be clearer. They are hardly legible now.
Authors: We lighter the bars in control condition in figure 1 and also improved the presentation of figure 1B and 1D.
Reviewer 2: Discussion is the weakest side of the job. The authors do not discuss with the results of other scientists, but quote a number of redundant information. Why they describe the course of photosynthesis? Why compare C3 plants with C4 plants when rice is a C3 plant? Why write about abscisic acid, if its level has not been studied by the authors? The whole fragment about ROS is unnecessary. You can mention these compounds, but they don't need to take up whole paragraphs. Figure 8 with description should be transferred to methods. The most interesting model of rice plants' response to the stresses studied (Fig. 9) should be much more precisely described and explained.
Authors: We improved the discussion with previous findings from other scientists. We also reorganized the text to not compare C3 and C4 plants. Sentence about abscisic acid was reduced only a few comment to complement the idea. The fragment related to ROS was excluded from the text. The final model was described and explained in more detail. Figure 8 was transferred to methods.
Reviewer 2: Final conclusion: The work must be rewritten, shortened, the results should be described separately from the discussion. Thanks to this, the work will be more transparent and legible. After the suggested corrections it can be resubmitted.
Authors: The manuscript was reformatted (rewritten and shortened) and the results were separated from the discussion. We improved the manuscript assuming all suggestions.

Round 2
Reviewer 2 Report
Manuscript was improved, but still needs some correction according to my suggestions:
- Abstract: line 18: ‘cultivated’ replace by ‘cultivars’
- 4. Methods: How could plants be measured after 24 hours of heat stress? They have grown? have decreased? You need to write how many days after the end of the stress were analyzed.
- 1. Results: line 202: ‘and’ replace by ‘while’
Line 212: ‘cultivated’ varieties replace by ‘cultivars’. Term ‘varieties’ is used in botanical systematic, while ‘cultivars’ in agronomy.
Lines 199; 268-271, 286-288, 350-358 transfer to Discussion
- Discussion: The discussion should avoid referring figures and tables. We will describe the results in the past tense and this form should be adhered to consistently, for example see lines 379-382.
Line 384 ‘cultivatedand’ please, improve it
‘cultivated rice genotype’ replace by ‘cultivars’
Lines 490-492; 517-527, 543-560 please delete.
Line 531; instead ‘here’ I suggest to write ‘in the present study’
Line 593 ‘increase’ add ‘of’ or ‘in’
- In my opinion English needs improvement
Author Response
Cover letter
Dear Reviewer 2,
Your comments were important to improve and ensure the quality of the manuscript. Below we list the reviewer’s comments and our responses.
Reviewer 2: Comments and Suggestions for Authors
Reviewer 2: Abstract: line 18: ‘cultivated’ replace by ‘cultivars’
Authors: Cultivated was replaced by cultivars.
Reviewer 2: Methods: How could plants be measured after 24 hours of heat stress? They have grown? have decreased? You need to write how many days after the end of the stress were analyzed.
Authors: Sorry, we mistake in way we written. Plants were measured 14 days after stress application. We wrote in methods this information.
Reviewer 2: Results: line 202: ‘and’ replace by ‘while’
Authors: The word and was replaced by while.
Reviewer 2: Line 212: ‘cultivated’ varieties replace by ‘cultivars’. Term ‘varieties’ is used in botanical systematic, while ‘cultivars’ in agronomy.
Authors: The term cultivated was replaced by cultivars along the text.
Reviewer 2: Lines 199; 268-271, 286-288, 350-358 transfer to Discussion
Authors: These sentences were transferred to Discussion section.
Reviewer 2: Discussion: The discussion should avoid referring figures and tables. We will describe the results in the past tense and this form should be adhered to consistently, for example see lines 379-382.
Authors: We deleted all figure and table references.
Reviewer 2: Line 384 ‘cultivatedand’ please, improve it ‘cultivated rice genotype’ replace by ‘cultivars’
Authors: ‘cultivated rice genotype’ was replaced by cultivars
Reviewer 2: Lines 490-492; 517-527, 543-560 please delete.
Authors: These sentences were deleted.
Reviewer 2: Line 531; instead ‘here’ I suggest to write ‘in the present study’
Authors: We changed ‘here’ by ‘in the present study’
Reviewer 2: Line 593 ‘increase’ add ‘of’ or ‘in’.
Authors: We inserted the word ‘in’
Reviewer 2: In my opinion English needs improvement
Authors: We checked all the manuscript, word-by-word and improved the English grammar which is highlighted with track changes.
